# Workplace Aesthetic Appreciation and Exhaustion in a COVID-19 Vaccination Center: The Role of Positive Affects and Interest in Art

**DOI:** 10.3390/ijerph192114288

**Published:** 2022-11-01

**Authors:** Fabrizio Scrima, Elena Foddai, Jean-Félix Hamel, Cindy Carrein-Lerouge, Olivier Codou, Benoit Montalan, Boris Vallée, Oulmann Zerhouni, Liliane Rioux, Pierenrico Marchesa

**Affiliations:** 1Department of Psychology, University of Rouen Normandy, 76130 Mont-Saint-Aignan, France; 2PLP Psicologi Liberi Professionisti, 90133 Palermo, Italy; 3Department of Psychology, University of Paris Nanterre, 92001 Nanterre, France; 4ARNAS Civico-Di Cristina-Benfratelli Hospital, 90127 Palermo, Italy

**Keywords:** aesthetic appreciation, positive affect, negative affect, art interest, exhaustion

## Abstract

Background: Recently, workers employed in vaccination points around the world have been subjected to very high workloads to counter the progress of the COVID-19 epidemic. This workload has a negative effect on their well-being. Environmental psychology studies have shown how the physical characteristics of the workplace environment can influence employees’ well-being. Furthermore, studies in the psychology of art show how art can improve the health of individuals. Objectives: The aim of this research was to test a moderated mediation model to verify how appreciation of workplace aesthetics can impact the level of exhaustion of staff working in a vaccination center, the mediating role of positive and negative affects, and the moderating role of interest in art. Methods: Data were collected from a sample of 274 workers (physicians, nurses, reception, and administrative staff) working in the same vaccination center in Italy. Participants answered a self-report questionnaire during a rest break. We used a cross-sectional design. Results: The results show that appreciation of workplace aesthetics impacts employees’ level of exhaustion. This relationship is mediated by positive and negative affects, and interest in art moderates the relationship between positive affects and exhaustion. Conclusions: These findings indicate the central role of workplace aesthetics in influencing healthcare workers’ well-being, and how interest in art can reduce exhaustion levels. Practical implications of the results are discussed.

## 1. Introduction

In January 2020, a worldwide public health emergency was declared following the outbreak of COVID-19 [1]. Despite efforts to reduce the spread of the virus, mass vaccination appeared to be the most effective strategy to eradicate the virus [2]. Accordingly, most countries have activated mass vaccination programs, setting up vaccination points in hospitals or specific vaccination sites [3]. Among other factors, the effectiveness of this procedure seems to be linked to the quality of the administration service [4]. However, vaccinating a large percentage of the population to achieve herd immunity places considerable emotional, cognitive, and physical demands on healthcare staff, and numerous studies have highlighted the impact of the pandemic on their psychological health [5]. According to Zhang et al. [6], medical staff show higher levels of stress, emotional exhaustion, and burnout compared to the general population or any other profession. A large number of studies have identified factors that protect health workers from stress [7], including environmental working conditions, and show how environmental satisfaction at work can impact their health [8]. For example, in a recent study, Brambilla et al. [9] showed that office location, natural or artificial light, and the indoor environment are factors that can positively or negatively affect physicians’ health. In the present paper, we explore additional variables that could promote employees’ health or be considered as protective factors against exhaustion. Exhaustion is one of the three dimensions of burnout, which is one of the most extensively studied work-related syndromes [10,11]. According to Maslach and Jackson [12], exhaustion is characterized by a weakening of one’s emotional resources, leading individuals to feel psychologically and emotionally exhausted. Laurence, Fried, and Slowik [13] found relationships between the physical characteristics of the workplace and exhaustion. Specifically, the aim of the present study is to identify the impact of appreciation of workplace aesthetics and interest in art [14] on the well-being of employees and how these factors can protect them against exhaustion. For example, Sirgy et al. [15] found a relationship between the aesthetics of the workplace and levels of satisfaction at work, while Kirillova et al. [16] found that front-of-house hotel employees exposed to art or luxury items had lower exhaustion levels than colleagues working in the back office. Furthermore, individuals with an interest in art are probably more open to aesthetic experiences. An aesthetic experience can be defined as a complex process associated with the cognitive and emotional responses of an individual exposed to artistic objects [17]; numerous studies [18] have shown the positive impact of aesthetic experiences on the health of individuals [19]. To the best of our knowledge, no research has investigated the impact of the perceived aesthetics of the workplace and artistic interests on the level of exhaustion of medical staff.

### 1.1. Workplace Aesthetic Appreciation and Exhaustion

Beauty can be considered as an ultimate value, traditionally studied by two schools of thought: formalism and contextualism [20]. Formalists argue that aesthetic experience depends on the intrinsic characteristics of the object and that beauty is a subjective, direct, and immediate reaction to it [21]. Contextualists argue that the aesthetic experience comes from the artist’s intention and contextual factors [22]. De Groot [23] found that triggering an aesthetic experience leads to a state of psychological well-being. Studies on the impact of an aesthetic environment on well-being can be traced back to Mintz [24], who showed empirically that individuals felt more energetic and had higher levels of well-being in a “beautiful” room than in an “ugly” room. Since then, numerous studies have investigated the beneficial effects of exposure to beautiful places on the well-being of individuals [25]. For example, living in a physically beautiful neighborhood improves the well-being of residents [26]. According to Strati [27], the physical environment of an organization can trigger an aesthetic experience. However, to date, few studies have investigated the impact of employees’ aesthetic appreciation of the organization on their well-being, stress, or burnout. Aesthetic appreciation of a place can be defined as an individual’s level of satisfaction with the aesthetic characteristics of a place. This appreciation results from the interaction between physical characteristics of the place and personal characteristics of the perceiver of a place (attitudes, values, beliefs) [28]. Moreover, aesthetic appreciation can be considered a part of the broader construct named environmental satisfaction, which refers to a person’s overall satisfaction with respect to their physical setting [29]. Kirillova et al. [16] found empirical evidence that employees exposed to design objects in a luxury hotel exhibit higher levels of well-being than colleagues working in the back office. Augustin [30] found that employees brought plants into their office to increase their aesthetic pleasure and that this had an impact on their well-being. In summary, while there have been no studies specifically investigating the relationship between workplace aesthetic appreciation and exhaustion among health workers, the findings of the studies mentioned above suggest that there is a relationship between aesthetic appreciation and level of exhaustion. Exhaustion can be considered the key dimension of burnout and is characterized by physical and mental fatigue and a feeling of emptiness of personal and professional resources [31]. In this sense, exhaustion could be considered the stress dimension of burnout [32]. We can therefore put forward the following hypothesis:

**H1.** 
*Aesthetic appreciation of the workplace is negatively related to exhaustion.*


### 1.2. The Mediating Role of Positive and Negative Affects

Aesthetic appreciation could modify the affective state of individuals. More generally, when people deliberately seek aesthetic experiences, positive emotional states are likely to occur [33]. Belke, Leder, and Augustin [34] found a positive association between positive affect and aesthetic appreciation. According to Proyer et al. [35], when people perceive beauty, they experience positive effects, while Moore and Marans [36] emphasized the restorative role of an environment that evokes a sense of beauty. Regarding the workplace, some physical characteristics (e.g., materials, colors, views, or lighting) are associated with positive evaluations of the work environment [37]. In a recent paper, Sander et al. [38] found that the sense of beauty related to the physical characteristics of the work environment are associated positively with positive affects and negatively with negative affects. The most commonly used construct for measuring positive and negative affects is that of Watson and Tellegen [39]. According to Watson et al. [40], positive and negative affects are two orthogonal dimensions of the affective experience; a high score on the Positive Affect Scale indicates an optimal state of energy, focus and enjoyable involvement, while a high score on the negative affect scale indicates a state of distress and unpleasurable engagement. Numerous studies have revealed the impact of positive and negative affects on the health of individuals. For example, Fredrickson, Tugade, Waugh, and Larkin [41] found that affect can protect against or trigger depressive states, and Steptoe, Wardle, and Marmot [42] noted that positive affects can lower cortisol levels that usually increase under severe stress. Wright and Cropanzano [43] also found that positive emotions are predictors of low levels of exhaustion, while Quaiser et al. [44] found that negative affects are predictors of exhaustion. The cited literature therefore allows us to put forward the following hypotheses:

**H2a.** 
*Workplace aesthetic appreciation is positively related to positive affects.*


**H2b.** 
*Workplace aesthetic appreciation is negatively related to negative affects.*


**H3a.** 
*Positive affects are negatively related to exhaustion.*


**H3b.** 
*Negative affects are positively related to exhaustion.*


### 1.3. The Moderating Role of Interest in Art in the Relation between Affects and Exhaustion

Hart [45] argued that interest in the arts can bring numerous benefits to physicians. Several studies have shown that being actively or passively interested in the arts (visual, musical, written) can impact on some traits that increase the ability of individuals to manage burnout; for example, Lampinen et al. [46] found that leisure activities, including passive interest in art, are a predictor of well-being in the elderly, and Mangione et al. [47] found that interest in the arts is a protective factor against burnout in medical students. Finally, in a study of internal medicine specialists, Orr et al. [48] observed that exposure to the visual arts reduced two dimensions of burnout, namely, emotional exhaustion and depersonalization. However, to date, there has been little research that explains why interest in art contributes to decreasing burnout. One possible explanation concerns the interaction between individual moods and art interests (Figure 1). Individuals with a strong interest in art will tend to fuel their interest through exposure to art. Art experts appreciate and understand works of art better than non-experts [49]. Furthermore, positive affects increase the appreciation of works of art [33], and this interaction could improve well-being [50]. Conversely, negative emotions associated with interest in art could reduce levels of appreciation and reduce levels of well-being. In accordance with this literature, we therefore hypothesize that:

**H4a.** 
*Interest in art moderates the relationship between positive affects and exhaustion.*


**H4b.** 
*Interest in art moderates the relationship between negative affects and exhaustion.*


## 2. Materials and Methods

### 2.1. Location, Participants, and Procedures

This research was carried out with a convenience sample of employees working in a vaccination center in Palermo (Italy) during the first and second administration of the COVID-19 vaccine. The vaccination center was located inside a building reorganized specifically for the city’s public vaccination service. This building was an open-plan with an area of about 5000 m^2^ organized into several semi-open stations with height walls. Each station was shared by physicians, for preliminary analysis, nurses for vaccine administration, and administrative staff for administrative paperwork. No artwork was present at the time the questionnaires were administered, however, different art exhibitions (e.g., photo exhibitions, or paintings) were planned to investigate the impact of artwork display on aesthetic appreciation of the workplace and exhaustion. The French law on biomedical research (Article L.1121-1-1 and Article R.1121-1 of the public health code) does not apply to this study, which was conducted in accordance with the American Psychological Association’s ethical principles and code of conduct for research with human participants [51]. Participants were recruited in the workplace on a voluntary basis. They were asked to complete a short questionnaire about the aesthetics of the interior and exterior design of their workplace and their subjective well-being. Data were collected through self-report questionnaires. Participants provided their informed consent after being assured that the data would be anonymous. According to Preacher, Rucker, and Hayes [52], a sample of at least 100 participants would be required to test our mediation model and detect an effect at *p* < 0.05 and a statistical power of over 0.80. Our sample thus comprised 275 participants, aged between 19 and 72 years (M = 36.27, SD = 12.43), 62% men and 38% women. Participants were asked to indicate the type of work they did in the center: 4.5% worked in reception, 57.9% in administrative offices, 14.7% were nurses, and 22.9% were doctors. To verify the statistical power of our sample we performed a post hoc power analysis using G*Power version 3.1 [53] with linear multiple regression: fixed deviation R^2^ of the model from zero. In line with Cohen [54], there were six predictors, 275 participants, a mean effect size of 0.15, and a level α = 0.05. The analysis reported a power of 0.99.

### 2.2. Tools

The administration of the questionnaire took about 15 min per person. The questionnaire included items providing personal data (gender, age, and type of work) and related to the variables studied.

Workplace aesthetic appreciation. The aesthetic appreciation of the workplace was evaluated using a single item: “how satisfied are you with the aesthetics of your workplace?”. Participants were asked to look around and think about the various spaces and places in the workplace and judge the aesthetic quality of the workplace. Participants responded on an 11-point scale from 0 “not at all” to 10 “totally”.

Positive and negative affects. The Italian version [55] of the international positive and negative affect schedule-short form [56] was administered to measure positive and negative affects. This scale consists of 10 items and is a short version of the scale originally developed by Watson, Clark, and Tellegen [40]. Five items measure positive affects (e.g., determined, attentive, alert), and five items measure negative affects (e.g., afraid, nervous, upset). Participants were asked to describe their mood at that precise moment. The response mode was a 5-point Likert scale from 1 (not at all) to 5 (very much). In the present study, we obtained α of 0.75 for positive affects and 0.83 for negative affects.

Exhaustion. Exhaustion was measured using six items from the Italian version of the organizational check-up system [57]. This scale has six items (e.g., “I feel fatigued when I get up in the morning and have to face another day on the job”; “I feel emotionally drained from my work”). The response mode was a 5-point Likert scale from 1 (Totally Disagree) to 5 (Totally Agree). In the present study, we obtained satisfactory internal consistency (α = 0.77).

Interest in art. We measured participants’ general interest in art using the Vienna Art Interest and Art Knowledge Questionnaire (VAIAK) [14]. An Italian version of VAIAK was created for the present paper using the back-translation technique and following the suggestions of Hambleton et al. [58]. This questionnaire comprises eleven items organized in two sets. The first set consists of seven items (e.g., I like to talk about art with others) rated on a 5-point Likert scale from 1 (Totally Disagree) to 5 (Totally Agree), and the second has four items (e.g., How often do you visit art museums and/or galleries?) rated on a 5-point Likert scale from 1 (Never) to 5 (Always). Confirmatory factor analysis showed a good fit index: χ^2^/df = 2.88, CFI = 0.96, NNFI = 0.94, RMSEA = 0.08. The internal consistency of the 11 items was satisfactory (α = 0.93).

### 2.3. Data Analysis

Data analyses were conducted using IBM SPSS statistics 20 software (SPSS Inc., Chicago, IL, USA). Descriptive statistics and the correlation matrix of the variables under study were calculated. After converting all the measures into Z scores, the PROCESS macro was used to test the hypotheses [59]. Specifically, we used model 14, with workplace aesthetic appreciation as an independent variable (IV), exhaustion as a dependent variable (DV), positive and negative affects as mediation variables (MeV), and interest in art as a moderation variable (MoV) on the relationship between positive and negative affects and exhaustion. Age and gender were introduced into the model as covariates. The significance of the effects of the moderated mediation model was assessed by calculating bootstrap confidence intervals. According to Hayes and Preachers [60], the effect is considered significant if the confidence interval does not include zero. For the significance of the mediation effect, the bootstrap confidence interval of the indirect effects was calculated. Conditional indirect effects were calculated for the significance of the moderation effect. Finally, we performed simple slope analyses to interpret moderation effects.

## 3. Results

### 3.1. Descriptive Statistics

Table 1 shows the means, standard deviations (SD), and bivariate correlations of the variables under study. Workplace aesthetic appreciation obtained a mean score of 5.60 (SD = 2.62), slightly lower than the theoretical average (6). There was a negative association between workplace aesthetic appreciation, gender, and age, with women (r = −0.20, *p* = 0.001) and older people (r = −0.15, *p* = 0.014) tending to be less appreciative of the workplace aesthetics. Workplace aesthetic appreciation was correlated positively with positive affect (r = 0.34, *p* < 0.001) and negatively with negative affect (r = −0.16, *p* = 0.009), art interest (r = −0.16, *p* = 0.01), and exhaustion (r = −0.37, *p* < 0.001). Positive affect obtained a mean of 3.46 (SD = 0.75) and was negatively correlated with gender (r = −0.16, *p* = 0.007), indicating that women tended to give a lower evaluation of their positive affective state than men. Positive affect was also negatively correlated with exhaustion (r = −0.37, *p* < 0.001). Negative affect (M = 1.38, SD = 0.56) was negatively correlated with age (r = −0.18, *p* = 0.002), with older people tending to give a lower evaluation of negative affects, which were positively correlated with exhaustion (r = 0.51, *p* < 0.001). Finally, interest in art (M = 2.67, SD = 0.93) correlated positively with age (r = 0.26, *p* < 0.001) and did not appear to be associated with exhaustion (r = 0.03, *p* = 0.586).

### 3.2. Hypothesis Testing

Table 2 shows the results of the moderated mediation model. Workplace aesthetic appreciation had a negative effect on exhaustion (B = −0.25, LL = −0.36 UL = −0.15), confirming our first hypothesis. It also had a positive impact on positive affect (B = 0.32, LL = 0.21 UL = 0.44), explaining 13% of the variance, and a negative impact on negative affect (B = −0.19, LL = −0.31 UL = −0.06), explaining 8% of the variance, confirming Hypotheses 2a and 2b, respectively. Positive affect negatively impacted exhaustion (B = −0.12, LL = −0.23 UL = −0.01), while negative affect positively impacted exhaustion (B = 0.47, LL = 0.37 UL = 0.57), confirming Hypotheses 3a and 3b, respectively. Workplace aesthetic appreciation, and positive and negative affects, explained 40% of the variance of exhaustion. Regarding the moderating effects of interest in art on the relationship between positive affect and exhaustion (H4a) and between negative affect and exhaustion (H4b), the only significant interaction was between positive affect and interest in art (B = −0.15, LL = −0.25 UL = −0.06), confirming hypothesis 4a but not hypothesis 4b. As shown in Figure 1, employees with high levels of interest in art who had low levels of positive affect reported higher levels of exhaustion than those who had high levels of positive affect.


**Table 2 ijerph-19-14288-t002:** Moderated mediation model.

	Positive Affect			Negative Affect			Exhaustion		
	B	SE	Bootstrap CI 95%	B	SE	Bootstrap CI 95%	B	SE	Bootstrap CI 95%
*Covariate*									
-Gender	−0.09	0.06	[−0.20, 0.02]	0.06	0.06	[−0.06, 0.18]	0.04	0.05	[−0.06, 0.13]
-Age	0.04	0.06	[−0.08, 0.15]	−0.23 ***	0.06	[−0.36, −0.11]	0.03	0.05	[−0.08, −0.13]
*Independent*									
-Workplace aesthetic appreciation	0.32 ***	0.06	[0.21, 0.44]	−0.19 **	0.06	[−0.31, −0.06]	−0.25 ***	0.06	[−0.36, −0.14]
*Mediator*									
-Positive affect							−0.12 *	0.06	[−0.23, −0.01]
-Negative affect							0.47 ***	0.05	[0.37, 0.57]
*Moderator*									
-Art interest							0.05	0.05	[−0.05, 0.15]
*Interaction*									
-PAF × ART							−0.15 **	0.05	[−0.25, −0.06]
-NAF × ART							0.02	0.05	[−0.07, 0.11]
R^2^		0.13			0.08			0.40	

Note: N = 274; * *p* < 0.05; ** *p* < 0.01; *** *p* < 0.001; PAF = Positive affect; NAF = Negative affect; ART = Art interest.

**Figure 1 ijerph-19-14288-f001:**
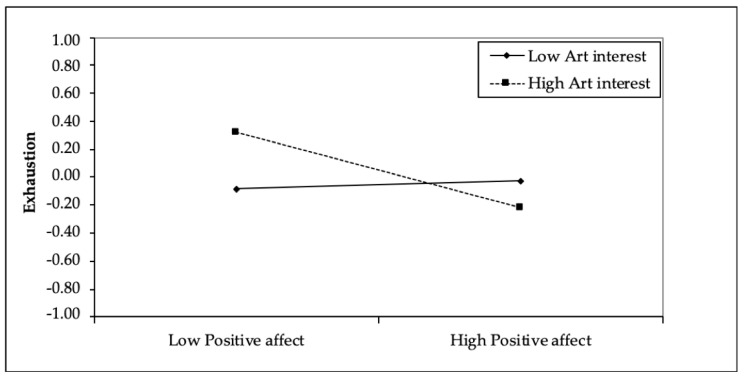
Simple slope analysis.

As indicated in Table 3, the moderation effect appears to be significant when interest in art is medium (Effect = −0.12, LL = −0.23 UP = −0.01) or high (Effect = −0.27, LL = −0.42 UP = −0.12). Table 4 confirms the existence of a moderated mediation effect only for positive affect (Effect = −0.05, LL = −0.09 UP = −0.02).

## 4. Discussion

Research indicating that the aesthetic appreciation of workplaces is a protective factor against exhaustion is very rare. Furthermore, most research has highlighted the positive impact of arts activities in reducing individual stress [50], but to the best of our knowledge, no research has ever investigated the role of employee artistic interests on exhaustion. This paper provides the first empirical evidence of the role played by workplace aesthetic appreciation on exhaustion, the mediating effect of positive and negative affects, and the moderating role of interest in art on the relationship between affect and exhaustion among workers in a vaccination center in Italy. While the exhaustion of medical staff may be due to excessive workload [61], ways of reducing it have been proposed by environmental psychologists [62]. For example, Amble [63] demonstrated that poor workplace design is associated with the level of employees’ stress, and that improving the design could have a positive impact on workers’ well-being. Other studies suggest that exposure to works of art can play a role in improving the well-being of individuals, particularly those with an interest in art [64].

Our hypotheses were tested with a moderated mediation model, with positive and negative affects as mediating variables in the relationship between aesthetic appreciation of the workplace and exhaustion, and interest in art as a moderating variable in the relationship between affects and exhaustion. Our results show that appreciation of workplace aesthetics negatively impacts exhaustion, confirming our first hypothesis, and they are in line with studies showing the restorative role of scenic beauty on the health of individuals [65]. According to Elsbach and Pratt [66], the attractiveness of the offices, furniture and building impacts the well-being of employees. People who work in aesthetically pleasing places report a better quality of life. Our results thus indicate that an aesthetically pleasing workplace can have a restorative effect on employee exhaustion. We observed a significant relationship between aesthetic appreciation of the workplace and positive and negative affects, confirming Hypotheses 2a and 2b, and in line with studies showing that the physical characteristics of the workplace have an impact on employee mood [67]. For example, Wasserman et al. [68] demonstrated how individuals react emotionally to physical cues in organizations, and Larsen et al. [69] found that introducing plants to make the office more attractive improves employee mood. These results indicate that working in an aesthetically pleasing environment can increase workers’ positive affects and reduce negative affects. Our results also confirm the hypotheses that positive and negative affects mediate the relationship between aesthetic appreciation and employee exhaustion (H3a and H3b), in line with other research. For example, Hwang et al. [70] found that positive and negative affects were associated with exhaustion, both before and after the COVID-19 pandemic; employees with high levels of positive affect manifested low levels of exhaustion, in contrast to those with high levels of negative affect who manifested high levels of exhaustion. This result is corroborated by a study by Qu, Yao, and Liu [71], drawing on COR theory [72,73]. The authors postulated that an increase reduces the resources available to cope with exhaustion. Conversely, upshifts in positive affects enable individuals to develop resources to cope with exhaustion, improving their state of well-being. Returning to the mediation effect, it is possible that being in an aesthetically pleasing environment would increase positive affects and reduce negative affects, with an impact on the resources available to cope with exhaustion. Finally, our results partially confirm Hypotheses 4a and 4b—that interest in art moderates the relationship between positive and negative affects and exhaustion, showing that only positive affects interact with interest in art to moderate the relationship between positive affects and exhaustion. Negative affects do not moderate this relationship. Forgas [74] found that affective states influence cognitive processing, while Leder et al. [33] observed that emotional states can influence the processing of art. More specifically, and in relation to COR theory [72,73], individuals with a positive affective state can feed their resources through frequent exposure to works of art, a typical behavior of people with a high interest in art. Conversely, people who have a high interest in art but a negative affective state may either decide not to look at art because they do not feel “well”, or, on the other hand, they may use art to improve their emotional state. This polarization could explain why interest in art does not moderate the relationship between negative affects and exhaustion.

The present study has several limitations. First, it is based on a convenience sample of people working in the same organization, and all the participants were volunteers, so we cannot rule out a self-selection bias. However, selecting participants from the same organization allowed us to control environmental variables that could have impacted the results. A second limitation might be that the administrations were performed at the workplace. The responses obtained may have been influenced by several disruptors. A third limitation concerns the cross-sectional design of the study, which made it impossible to establish cause and effect relationships, and the hypothetical directions of the effect were based on the scientific literature. Further longitudinal studies would help demonstrate the directionality of the effects. In addition, workplace aesthetic appreciation was measured with a single item, created for the purposes of the research investigating the level of satisfaction about workplace aesthetics. Future research should measure workplace aesthetic appreciation more appropriately. However, the authors relied on suggestions from the literature in environmental psychology that measures aesthetic appreciation of a place by asking the level of satisfaction [75]. Finally, our study did not consider other personality or situational variables that could have an impact on exhaustion (see [76,77]).

Despite these limitations, our results suggest two possible approaches that organizations could take to reduce the level of employee exhaustion and stress: first, improving the aesthetics of the working environment, and second, developing their employees’ artistic interest. Regarding the former, improving workplace aesthetics does not require large sums of money; for example, introducing plants and flowers [78], allowing employees to personalize their workspace with personal objects [79], or putting up art posters in the workplace [80]. Regarding the second approach, studies have shown that art appreciation courses for medical staff can reduce burnout levels [46,47]. It would thus be interesting for managers to create partnerships between hospitals and museums or other art institutions to develop the art appreciation of medical staff in order to reduce exhaustion levels.

## 5. Conclusions

Drawing on COR theory, this study investigated two aspects that can impact the levels of exhaustion of medical staff engaged in a mass vaccination program against COVID-19. The results show, for the first time, that aesthetic appreciation of the workplace has an impact on the level of exhaustion of medical personnel, that positive and negative moods can mediate this relationship, and that interest in art moderates the relationship between positive moods and exhaustion. They also show how managers can intervene in two specific areas to promote the well-being of medical staff: first by enhancing the attractiveness of the workplace, and secondly by developing the artistic interests of the staff. However, these relationships require further investigations to overcome the limitations of the study.

## Figures and Tables

**Table 1 ijerph-19-14288-t001:** Means, SDs, correlations, and Alphas on the diagonal.

		Min	Max	Mean	SD	1	2	3	4	5	6	7
1	Gender	-	-	-	-	-						
2	Age	19	72	36.27	12.43	0.01	-					
3	Workplace aesthetic appreciation	0	10	5.60	2.62	−0.20 **	−0.15 *	-				
4	Positive affect	1	5	3.46	0.75	−0.16 **	−0.01	0.34 **	0.76			
5	Negative affect	1	5	1.38	0.56	0.09	−0.18 **	−0.16 **	−0.15 *	0.83		
6	Art interest	1	5	2.67	0.93	0.03	0.26 **	−0.16 **	0.13 *	−0.07	0.93	
7	Exhaustion	1	5	2.07	0.76	0.14 *	−0.00	−0.37 **	−0.27 **	0.51 **	0.03	0.77

Note: N = 274; * *p* < 0.05; ** *p* < 0.01.

**Table 3 ijerph-19-14288-t003:** Conditional effect of the focal predictor.

Art Interest	Effect	SE	CI 95%
−1 SD	0.03	0.07	[−0.11, 0.17]
Mean	−0.12 *	0.06	[−0.23, −0.01]
1 SD	−0.27 ***	0.08	[−0.42, −0.12]

Note: * *p* < 0.05; *** *p* <0.001.

**Table 4 ijerph-19-14288-t004:** Conditional indirect effect.

**Through Positive Affect**			
**Art Interest**	**Effect**	**SE**	**CI 95%**
−1 SD	0.01	0.02	[−0.03, 0.05]
Mean	−0.04	0.02	[−0.08, −0.01]
1 SD	−0.09	0.03	[−0.15, −0.04]
Index of moderated mediation	−0.05	0.02	[−0.09, −0.02]
**Through Negative Affect**			
**Art Interest**	**Effect**	**SE**	**CI 95%**
−1 SD	−0.08	0.03	[−0.15, −0.03]
Mean	−0.09	0.03	[−0.15, −0.03]
1 SD	−0.09	0.03	[−0.16, −0.03]
Index of moderated mediation	−0.00	0.01	[−0.02, 0.02]

## Data Availability

The data presented in this study are available on request from the corresponding author.

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
