# Peer review of "Workplace Aesthetic Appreciation and Exhaustion in a COVID-19 Vaccination Center: The Role of Positive Affects and Interest in Art"

_ijerph, 2022, doi:10.3390/ijerph192114288_

Round 1

Reviewer 1 Report

The research is timely and the manuscript well-written.

The term "workplace aesthetic appreciation" needs to be defined. It is not a construct with which I am familiar. How does this differ from workplace environmental satisfaction, which frequently appears in the literature? The one item used to measure WAA ask specifically about SATISFACTION with aesthetics in the workplace.

Internal validity: Given that just one item is used to assess WAA and it does not appear to have been previously validated, I am concerned that WAA is not actually being evaluated properly. At a minimum, this should be addressed in the limitations. Validity is my biggest concern with this manuscript.

Workplace aesthetics: Although it may be irrelevant whether or not the workplace environment would generally be considered pleasing or not, a brief description of the physical environment would be helpful - specifically with respect to the presence of art, since art appreciation is an important construct in this investigation. Otherwise, it is unclear why this specific site was chosen to examine the moderating effect of art appreciation.

Does the fact that more than half of participants work in administrative offices and are not clinicians potentially skew the findings? How does their workplace environment differ from that of the clinicians? Did you investigate differences between admin/reception and clinicians, which may be important given the differences in the types of work that they do and the levels of workplace exhaustion?

Conclusions: The mention of human resources managers seems too narrow, as these are not typically the people responsible for improving the physical environment of a healthcare workplace, at least not in the US.

Author Response

Dear colleague,

Thank you for giving us the opportunity to review our paper. The concerns raised by the referees are appropriate and constructive. Attached, you will find the list of concerns raised by the referees and our responses in red.

Reviewer 2 Report

Please explain at paragraph 1.1 what exhaustion is, making references to the the previous literature;

Please explain if the administred scales were already validatated in Italy;

Please, explain what s the gap in the literature that you paper aims to address and why this is relevant to the existing knowledge;

Please comment in the limits paragraph about the results reached in the participants, if possible administration of the same instruments might have offered different findings 

Author Response

(The authors gave the same response as above.)
